# Global reporting and underreporting of occupational diseases: A systematic review

**Levina Chandra Khoe** [ID][1,2]*, **Siti Rizny Fitriana Saldi**[3], **Marsen Isbayuputra**[2],
**Muchtaruddin Mansyur** [ID][2‡], **Virginia Wiseman**[4‡], **Augustine Asante**[1]

**1** School of Population Health, University of New South Wales, Sydney, Australia, **2** Department of
Community Medicine, University of Indonesia, Jakarta, Indonesia, **3** Center for Clinical Epidemiology and
Evidence-Based Medicine, Cipto Mangunkusumo Hospital, Jakarta, Indonesia, **4** Kirby Institute, University
of New South Wales, Sydney, Australia

☺ These authors contributed equally to this work.
‡ These authors contributed equally to this work.
* l.khoe@unsw.edu.au; levina.chandra01@ui.ac.id

## Abstract

### Background

Disease reporting is often unreliable and faces many challenges, making it difficult
to estimate the true burden of occupational diseases, defined as any disease that
is caused by work activities or environment. This study aimed to assess the global
reporting and underreporting rate of occupational diseases, and to identify the factors
affecting the underreporting of occupational diseases.

### Methods

Following the Preferred Reporting Items for Systematic Reviews and Meta-Analysis
(PRISMA) guidelines, this study searched Medline (PubMed), CINAHL, EMBASE,
Scopus, Web of Science, WHO Institutional Repository for Information Sharing
(IRIS) database, Dimensions, and Google Scholar in September 2024. Search terms
related to reporting and underreporting of occupational diseases or illnesses were
used. The selected records were screened, and data extracted using the Covidence
software tool. Screening and quality assessment were conducted by two independent researchers and finalized by a third researcher. The quality of the evidence was
assessed with the Mixed Methods Appraisal Tool. This study is registered on PROSPERO, number CRD42023417814.

### Results

A total of 127 studies from 29 countries were identified, all coming from high-income
and upper-middle-income countries. The incidence rate of occupational disease varied widely, ranging between 1.71 to 1,387 per 100,000 employees yearly. The highest
number of annual cases was reported in the agricultural sector (ranging from 33 to

org/10.1371/journal.pone.0345318

of Science, UNITED KINGDOM OF GREAT
BRITAIN AND NORTHERN IRELAND

**Peer Review History:** PLOS recognizes the
benefits of transparency in the peer review
process; therefore, we enable the publication
of all of the content of peer review and
author responses alongside final, published
articles. The editorial history of this article is
available here: https://doi.org/10.1371/journal.
pone.0345318

**Data availability statement:** All relevant data are within the manuscript and its Supporting Information files.

**Funding:** This work was supported by the Indonesian Endowment Fund for Education (LPDP), Ministry of Finance, Republic of Indonesia (Grant Number: S-2431/LPDP.4/2022 to LK). The funder had no role in study design, data collection and analysis, decision to publish, or preparation of the manuscript.

**Competing interests:** The authors have declared that no competing interests exist.

6,431), followed by the health sector (146–5,508), and then the construction sector (264). Two studies evaluated rates of underreporting, which varied from 50% to 95%. The main factor contributing to underreporting was employee concerns about job security.

## Conclusions

The results reveal a significant gap in the reporting of occupational diseases among high-income and low-middle-income countries. Variations in reporting mechanisms across countries were also identified. Our findings highlight the need to establish a national system for reporting occupational diseases that engages employers, employees, and healthcare providers.

## Introduction

In 2016, the World Health Organization (WHO) and the International Labour Organization (ILO) produced the first joint estimates of the work-related burden of disease and injury. They reported a total of 1.9 million deaths and a loss of 89.7 million disability-adjusted life years (DALYs) attributable to 19 occupational risk factors [1]. A more recent study by Takala et al estimated a higher number of occupational diseases, with 2.58 million deaths in 2019 [2]. Occupational disease is defined as any disease that occurs because of exposure to hazards from work activity or environment [3]. If workers are exposed to any of these hazards in the workplace, such as noise, vibration, chemical agents, biological agents, awkward postures, or long work hours, they are at risk of developing occupational diseases [4]. There are two main criteria that differentiate occupational diseases from non-occupational diseases, i.e., the hazard exposure at workplace significantly increases the risk of the disease, and the disease prevalence among workers is higher than the general population [4]. The impact of occupational diseases is not just limited to the health and quality of life of workers but can also lead to significant economic burden [5].

To prevent occupational diseases, obtaining correct information on their prevalence is necessary. According to a global survey conducted by the WHO, about 93.5% of countries globally collect data on occupational diseases and 68.5% of these maintain a national registry of occupational diseases and accidents [6]. The same survey also shows that about 87.2% of high-income countries (HICs) have national registries, and around 40.8% of low- and middle-income countries (LMICs) do not have these registries. Even where national registries exist, the level of registration is reported to be low, especially in LMICs [6]. Concerns about incomplete registry data especially around occupational risk factors has also been raised [7].

There are recognized challenges associated with the collection of occupational diseases data in terms of population coverage, source of information, and coordination between multiple authorities [7]. Firstly, the population covered by reporting systems mainly includes employees working in the formal sector, excluding the self-employed, part-time workers, casual workers, seasonal workers, and those in

micro to small enterprises [8]. These gaps in population coverage result in underreporting, especially in LMICs, where more than two-thirds of the population works in the informal sector [9]. Secondly, to diagnose occupational disease, physicians require robust evidence on the possible occupational origins and their level of exposure [10,11], which is typically sourced from employers [12]. Insufficient training in occupational medicine for physicians has also been identified as one of the factors contributing to the under-diagnosis of occupational diseases [13–15]. Apart from human factors, some occupational diseases require a long latency period before the appearance of first symptoms. For instance, the latency period from exposure to carcinogenic substances for bladder cancer is about 14 years, which makes it difficult to establish the link between exposure and disease [16]. Lastly, the responsibility to collect data on occupational diseases may be divided among different organizations. In many countries, the Ministry of Health and the Ministry of Labour often require this information for disease surveillance purposes and for developing occupational health programmes. Employers and insurance agencies need the information for workers' compensation schemes [17]. Each organization may use different reporting mechanisms, which could increase the complexity and fragmentation of documenting occupational diseases [6].

Occupational disease reporting systems are important sources of information for understanding disease patterns among workers in a country and for developing effective prevention programmes. However, statistics on occupational diseases are often unreliable and remain scarce in many countries, particularly in LMICs, where many workers have a higher risk of developing occupational diseases [18]. While all cases of occupational disease must be reported by employers, employees, and/or physicians, many employees are unaware that their disease may be caused or worsened by the work environment [19,20]. Some may be aware but choose not to report for fear of potential repercussions, including losing their jobs [21]. Factors that contribute to underreporting of occupational diseases might also originate from employers who may not consider the employee's case as work-related. Additionally, there is a lack of enforcement of occupational health and safety regulations in many countries. A lack of strictly enforced sanctions on employers who fail to report confirmed or suspected occupational-related diseases to authorities has been documented in many countries [21–24].

Occupational diseases are often described using the "icebergs phenomenon" [25]. This means that most cases may be invisible or mistakenly diagnosed as non-occupational diseases. Thus, the number of cases reported in many countries may be just the tip of the iceberg. While studies have documented the burden of occupational diseases, there has been limited effort to explore the global disease pattern and identify its reporting mechanisms. Underreporting of occupational diseases limits our understanding of the true burden of these diseases, leading to misallocation of resources and potentially ineffective prevention efforts. To our knowledge, no systematic review exists on the global reporting of occupational diseases. This systematic review aims to fill that gap by bringing together evidence on the reporting of occupational diseases in all countries. The review covers all types of occupational diseases. Occupational injuries or accidents are not included because they are more easily recognized and better reported compared with most diseases. Our goal is to systematically review the reporting and underreporting pattern of occupational diseases based on countries' income status, industrial development, and types of occupational diseases. We also aim to identify the factors affecting the underreporting of occupational diseases.

## Methods

The protocol for this review was prepared according to the Center for Reviews and Dissemination (CRD) guidelines and registered on PROSPERO, under registration number CRD42023417814. The selection of studies followed the Preferred Reporting Items for Systematic Review and Meta-Analysis-Protocols (PRISMA) 2020 guidelines. The quality of the studies were assessed using the Mixed Methods Appraisal Tool (MMAT) [26]. This review is part of LCK doctoral study, which received ethics approval from the University of New South Wales (HC220796) and the Ethics Committee at University of Indonesia (KET-189/UN2.F1/ETIK/PPM.00.02/2023).

## Eligibility criteria

Studies were eligible for inclusion if they examined the reporting or underreporting of occupational diseases at the country level. Eligible studies included cohort studies, cross-sectional studies or reports from national registries, national surveillance systems, workers' compensation schemes, or national voluntary reporting schemes. As this review focused on the reporting system at the national level, studies at the district or province level were excluded. We included studies that provide real world data and excluded purely modeling studies. All types of occupational diseases were included; however, occupational injuries or accidents were excluded. Review articles, editorials, guidelines, case reports, and case series were also excluded, as our focus was on empirical studies. Qualitative and mixed-methods studies were included to identify the factors underlying problems of underreporting. There were no limitations on the date of publications and language. For non-English language articles, Google Translate was used to screen titles and abstracts. These articles were kept in a separate folder but not included in the analysis, to avoid language bias when appraising the articles.

## Outcome

Outcome measures include the number of reported occupational disease cases, the rate of reporting occupational diseases, the rate of underreporting or misreporting, the number of occupational disease claims, the number of cases reported by physicians, employees or employers. The underreporting rate is defined as the ratio between the number of non-reported cases and the total number of cases (reported and not reported). The rate of misreporting refers to the ratio between the number of falsely reported cases and the total number of cases. Reported cases were classified according to the countries' income status, industrial development, and types of occupational diseases.

## Data sources and search strategy

Searches were conducted for the following electronic databases: Medline (PubMed), CINAHL, EMBASE, Scopus, and Web of Science. Searches for eligible grey literature were carried out in the WHO Institutional Repository for Information Sharing (IRIS) database, Dimensions, and Google Scholar. Additionally, the reference lists of relevant articles were screened for titles and abstracts that include key terms.

We explored different possible terms related to reporting and underreporting of occupational diseases/illnesses including "report*", "underreport*", "misreport*", "surveillance", and "capture-recapture", and combined them with terms for occupational diseases ("occupational disease*", "occupational illness*", "work-related disease", "work-related illness"). The search strategy included a combination of Medical Subject Heading (MESH) terms and free text terms. These terms were combined with 'OR' and 'AND' Boolean operators. The search in all databases was initially performed on May 2nd, 2023, and updated on September 30th, 2024, in as described in details in S1 Appendix.

## Study selection

Studies obtained from different data sources were combined and duplicate records were removed using the Covidence systematic review software package. All records identified in the search were initially screened based on titles and abstracts. Then, we assessed the full text of selected studies according to the eligibility criteria. Study selection was performed by two independent investigators (LK and SR or LK and MI). Any disagreements were resolved through discussion with a third reviewer (SR or MI). The third reviewer was a person not involved in the study selection process.

## Data extraction

A standardized form was developed for data extraction based on the review questions. Extracted information included basic study details such as study design, country of origin, study setting, year or timeframe for data collection, participant employment characteristics (industrial sector, job type), and the outcome data (e.g., number of cases reported, reporting

rate). Additionally, the countries where the studies were conducted were stratified by income status and region according to the World Bank classification [27]. One reviewer (LK) extracted the data, while a second reviewer (SR or MI) checked the data for accuracy and completeness. Any discrepancies between reviewers were resolved through consensus, or by involving a third reviewer (SR or MI) to settle the disagreement in the data extraction process. Where essential information was unclear or missing, the authors were contacted for clarification. Completed data extraction forms were uploaded to the university's (University of New South Wales) One Drive account, which was accessible only to the reviewers.

## Quality assessment

At least two reviewers (LK and SR or LK and MI) assessed and appraised the methodological quality of the studies independently using Mixed Methods Appraisal Tool (MMAT) [26]. The types of studies assessed using MMAT include qualitative studies, quantitative randomized controlled trials, quantitative non-randomized studies, quantitative descriptive studies, and mixed-method studies. Five criteria were used to assess the overall risk of bias for each study type. Each criterion is given a rating of 'yes', 'no', or 'can't tell'. For every 'yes' answer, the study was given a score of between 20 and 100, with 20 being the lowest and 100 the highest. If different results were obtained, the assigned investigators discussed until an agreement was reached. If no agreement was reached, a third investigator (SR or MI) was consulted to obtain an agreement.

## Analysis and data synthesis

The articles were categorized based on their content. Articles that included the number of reported cases or incidence rate of occupational diseases were grouped into one folder and analysed to address the primary objective, i.e., to estimate the global reporting and underreporting rate of occupational diseases. Descriptive data from each reviewed study were presented as narrative text or in tables. Due to the diversity in the characteristics of the studies, conducting a meta-analysis was not possible. Thus, a narrative synthesis was performed for this systematic review. A summary table showing the number of cases per year and incidence of occupational diseases per 100,000 employees was presented. Where the incidence was originally reported using a different denominator (e.g., per one million or one thousand employees), the number was adjusted to a denominator of per 100,000 employees. The results were also classified based on the type of occupational diseases in each region, and the type of occupational diseases in each industrial sector. In every study, we calculated the number of annual cases for each type of occupational disease by adding up the total cases and dividing by the total number of years covered by the study to obtain the average number of cases per year. The average number of cases per year in each study were summed up and then divided by the total number of studies. For studies discussing the factors contributing to the underreporting of occupational diseases, these articles were separated into a separate folder and analysed thematically.

## Results

### Characteristics of included studies

From the searched databases, we identified 15,814 records. Of these, 7,038 duplicates were removed. After title and abstract screening, the remaining 422 full-text articles were assessed for eligibility. Of these 422 articles, 21 studies were in other languages; 3 German, 2 French, 4 Spanish, 2 Danish, 3 Polish, 2 Italian, 2 Norwegian, and 3 Chinese. A total of 295 studies were excluded from the 416 because they did not meet the inclusion and exclusion criteria. That left a total of 127 articles to be considered for final review, including 123 quantitative studies and 4 qualitative studies. The study selection process is shown in Fig 1. These 127 studies covered 29 countries spread across all six geographical regions of the world (Africa, Asia, Australia and Oceania, Europe, Northern America, and Latin America). The first study was published in 1990 and the most recent in 2024. While only 15 studies were published in the first 10 years between 1990 and 2000; the

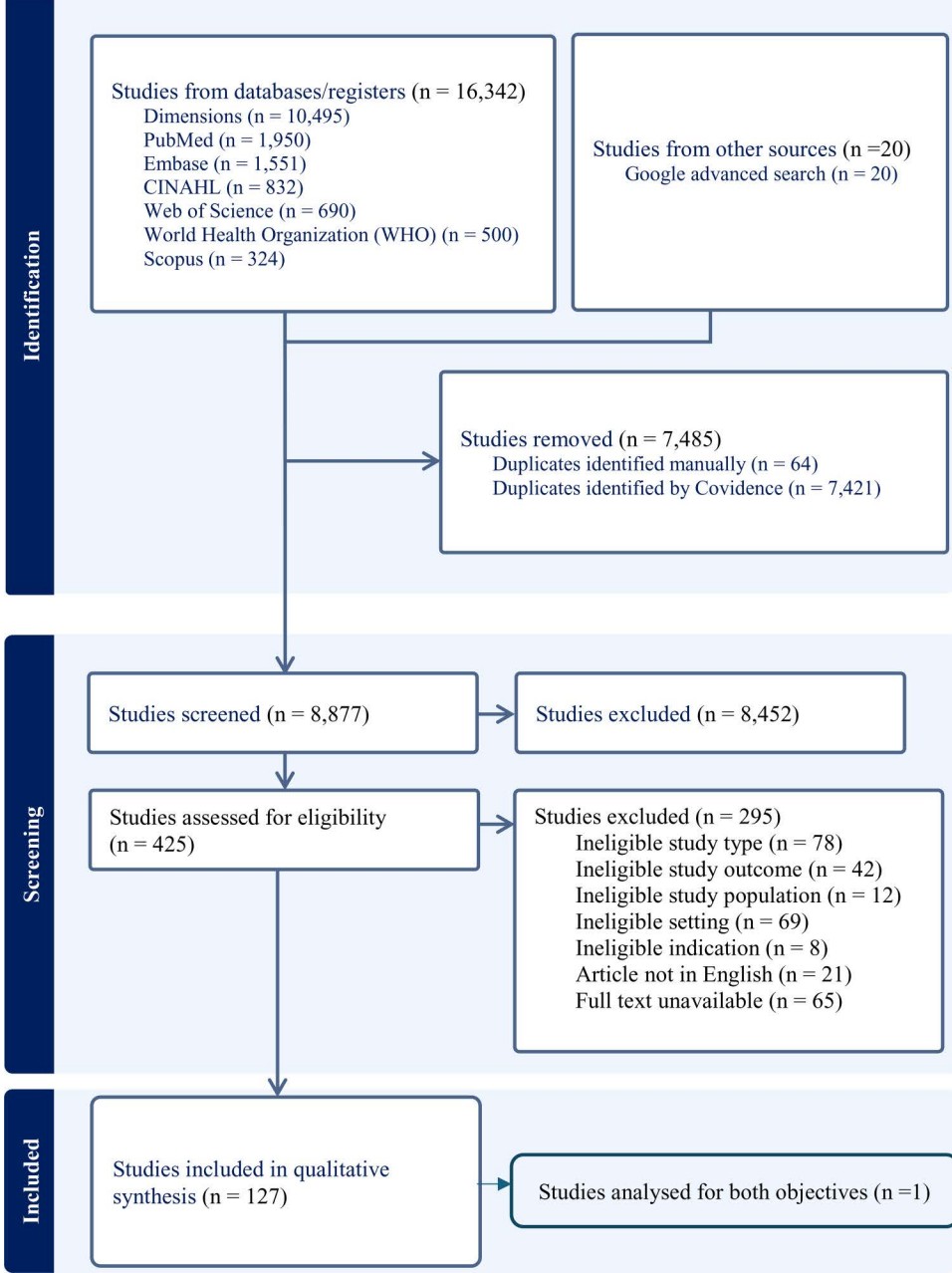

**Fig 1. PRISMA flow diagram of study selection.**

number of published articles remained stagnant each year, at around 6 studies annually. The characteristics of all eligible articles are presented in Table 1.

The country with the highest number of publications on occupational diseases was the United Kingdom (UK), which published 29 articles. Other top-producing countries included South Korea and Finland, contributing 9 and 8 articles respectively. Denmark, Australia, and Canada were the first countries to publish occupational disease studies between 1990 and 1996. All the articles included in this review were published in high-income and upper-middle-income countries.

**Table 1. Summary characteristics of all studies.**

| Characteristics | n of articles |
|---|---|
| Year published | |
| 2019–2024 | 30 |
| 2013–2018 | 30 |
| 2007–2012 | 32 |
| 2001–2006 | 20 |
| < 2001 | 15 |
| Region | |
| Europe | 98 |
| Asia | 16 |
| North & South America, Canada | 9 |
| Australia | 2 |
| Africa | 2 |
| Industrial sector | |
| All sectors | 92 |
| Agriculture | 7 |
| Mining and quarrying | 3 |
| Manufacturing | 6 |
| Construction | 3 |
| Healthcare | 10 |
| Service | 2 |
| Others | 5 |
| Type of diseases | |
| General / non-specific occupational diseases | 36 |
| Skin diseases | 20 |
| Cancer | 17 |
| Respiratory diseases | 13 |
| Occupational asthma | 10 |
| Musculoskeletal disorders | 9 |
| Infectious diseases | 8 |
| Asbestos-related diseases | 7 |
| Mental illnesses | 3 |
| Hearing damage/loss | 2 |
| Poisoning | 1 |
| Chronic condition | 1 |
| Assessed more than one disease | 7 |
| Data sources* | |
| Physician reports | 43 |
| Occupational health registries | 40 |
| Workers' compensation claims | 23 |
| Others (cancer registries, labour surveys) | 17 |

*exclude studies focusing on factors contributing to underreporting

Across geographical regions, Europe contributed the highest number of publications [98] while the regions of Latin America, Africa, and Australia had the least number of publications; only two each. Of the 123 articles that assessed the rate of occupational disease reporting, 43 studies were based on physician reports, 40 studies used data sources from occupational health registries, 23 studies used data from workers' compensation claims, and the rest gathered data from a variety of sources including cancer registries and labour surveys.

A total of 91 studies reported on specific occupational diseases, and the most commonly studied conditions were skin diseases (20 studies), cancer (17 studies), respiratory diseases (13 studies), occupational asthma (10 studies), musculoskeletal disorders (9 studies), infectious diseases (8 studies), asbestos-related diseases (7 studies), mental illnesses (3 studies), hearing damage or loss (2 studies), poisoning (1 study), and other chronic conditions (1 study). About 7 studies

investigated more than one type of occupational disease. In terms of industry, the health sector had the highest number of studies [10], followed by construction and/or manufacturing [9], agriculture [7], mining and quarrying [3], service [2], and other industrial sectors [5]. We present a summary characteristics of all studies in Table 1 and a detailed characteristics of each study in Table 2.

## Quality of studies

The majority of the 127 studies (n = 121) used a quantitative descriptive research design, followed by qualitative design (n = 4), and quantitative non-randomized design (n = 2). The quality of the studies was appraised using MMAT with the highest score of 100. Studies that obtained high scores in terms of quality were qualitative studies, with 50.0% (n = 2) scored 100 and 50.0% (n = 2) scored 80. For these qualitative studies, a perfect score [100] indicates that a paper has met all the quality criteria. Such excellence is reflected in a study design that is fully aligned with the research questions and supported by data collection methods that address those questions comprehensively. These papers also demonstrate strong analytical integrity, ensuring that findings arise directly from the evidence and that all interpretations are substantiated by the data. Moreover, they maintain a coherent and logical connection between the data sources, data collection processes, analytical procedures, and final interpretations. Only two studies were rated as having moderate quality (score = 80), as they provided insufficient information to fully support their reported findings.

Among the quantitative descriptive studies, most (82.1%) had an MMAT score of 80 or higher (n = 101). Only 7 studies (5.7%) scored 40 and 15 studies (12.2%) scored 60. Meanwhile, two quantitative non-randomized studies received a of scored 60. Scores in this range generally reflect limited clarity and inadequate explanation of core methodological components. In particular, the absence of detailed information on measurement methods, sampling strategies, and population coverage made it impossible to fully assess the rigor of these studies. These omissions may also limit the generalizability of their findings.

The remaining results are presented according to the study objectives, type of occupational disease, and type of industrial sector. We separated those articles to be analysed for the primary objective, i.e., to estimate the global reporting and underreporting rate of occupational diseases, and the secondary objective of identifying the factors contributing to the underreporting of occupational diseases. Eight articles were assessed for the secondary objective, and the rest were analysed for the primary objective. One article was assessed for both objectives. Of the 119 articles assessed for the primary objective, only two examined underreporting patterns in occupational diseases. Ninety-three [93] of the 119 studies examined only a specific occupational disease without considering the type of industrial sector. Meanwhile, there were 15 articles which discussed all types of occupational diseases among the general working population. Fig 2 below describes the number of studies in each category.

## Reporting patterns for occupational disease

Data from 15 studies indicate that the number of cases of occupational diseases among the general working population globally ranged from 34 to 37,927 per year with an incidence rate of between 1.71 to 1,387 per 100,000 employees per annum. Almost all the cases were reported in the European region, except for two cases, one in Taiwan and the other one in Turkey. The highest rate of occupational disease incidence, as reported by general practitioners between 2006 and 2009, was recorded in the UK. The lowest incidence was recorded in Greece, and it was based on data from workers' compensation claims. Three specific sectors: agriculture, healthcare, and construction, reported all types of occupational diseases from skin diseases to cancer. The reported annual cases of occupational diseases ranged between 33–6,431 in the agricultural sector; 146–5,508 in the healthcare sector, and 264 in the construction sector. Table 3 summarises the number of reported cases and incidence rate of occupational diseases in general and specific working populations.

**Table 2. Characteristics of each included study.**

| No | Author | Year | Category of study design | Time period of data collection | Type of illness | Country | Industrial sector |
|----|--------|------|--------------------------|-------------------------------|-----------------|---------|-------------------|
| 1 | Murphy PL [28] | 1999 | Quantitative descriptive | 1987-1995 | Occupational low back pain | United States | All |
| 2 | Kraut A [29] | 1994 | Quantitative descriptive | 1989-1991 | All | Canada | All |
| 3 | Nordman H [30] | 1999 | Quantitative descriptive | 1990-1995 | Occupational asthma | Finland, Sweden, UK, Canada, US | Food, manufacturing, electronic, wood working industry |
| 4 | Pelclova D [31] | 2007 | Quantitative descriptive | 1991-2005 | Mesothelioma | Czech Republic | Asbestos exposed industries |
| 5 | Ross DJ [32] | 1998 | Quantitative descriptive | 1997 | work related and occupational respiratory diseases | UK | Latex-related industries |
| 6 | McDonald JC [33] | 2006 | Quantitative descriptive | 1996-2001 | work related skin diseases | UK | All |
| 7 | Chen H [34] | 2013 | Quantitative descriptive | 2001-2010 | Coal miners' occupational disease | China | Coal mining industries |
| 8 | Toren K [35] | 1996 | Quantitative descriptive | 1990-1992 | Occupational asthma | Sweden | All |
| 9 | McDonald JC [36] | 2005 | Quantitative descriptive | 1992-2001 | Acute work related respiratory diseases | UK | All |
| 10 | Sirajuddin H [37] | 2001 | Quantitative descriptive | 1997-1998 | Lung diseases, occupational dermatosis, poisoning | Malaysia | All |
| 11 | Moldovan HR [38] | 2017 | Quantitative descriptive | 2013-2014 | Occupational skin diseases | 22 Eastern European countries | All |
| 12 | Baur X [39] | 2005 | Quantitative descriptive | 2003 | Non-malignant occupational airway diseases | Germany | All |
| 13 | Ding Q [40] | 2013 | Qualitative | 2000-2010 | Pneumoconiosis, poisonings | China | All |
| 14 | Machovcova A [41] | 2013 | Quantitative descriptive | 1997-2009 | Occupational skin diseases | Czech Republic | Healthcare industries |
| 15 | Turner S [42] | 2005 | Quantitative descriptive | 2000-2003 | Work-related infectious diseases | UK | All |
| 16 | Walsh L [43] | 2005 | Quantitative descriptive | 2002-2003 | Occupational diseases | UK | Health and social work sectors |
| 17 | Pal TM [44] | 2009 | Quantitative descriptive | 2001-2005 | Occupational skin diseases | Netherlands | All |
| 18 | Samant Y [45] | 2020 | Quantitative descriptive | 2007-2016 | Occupational diseases | Norway | Agriculture |
| 19 | Soo Oh S [46] | 2010 | Quantitative descriptive | 1992-2006 | Occupational asthma | South Korea | All |
| 20 | van der Molen HF [47] | 2012 | Quantitative descriptive | 2009-2013 | Occupational diseases | Netherlands | Economic sector |
| 21 | Burnett CA [48] | 1998 | Quantitative descriptive | 1993 | Occupational dermatitis | United States | Private industries |
| 22 | Aalto-Korte K [49] | 2020 | Quantitative descriptive | 2005-2016 | Occupational skin diseases | Finland | All |
| 23 | Spiewak R [50] | 2003 | Quantitative descriptive | 1991-1999 | Occupational dermatoses | Poland | Agriculture |
| 24 | Karjalainen A [51] | 1997 | Quantitative descriptive | 1960-1995 | Mesothelioma | Finland | All |

*(Continued)*

| No | Author | Year | Category of study design | Time period of data collection | Type of illness | Country | Industrial sector |
|----|--------|------|--------------------------|-------------------------------|-----------------|---------|-------------------|
| 25 | Ahn Y [52] | 2008 | Quantitative descriptive | 1998-2004 | Occupational infectious diseases | South Korea | Healthcare industries |
| 26 | Cherry NM [53] | 2006 | Quantitative descriptive | 1996-2001 | Work-related stress and mental ill-health | UK | All |
| 27 | Meyer JD [54] | 2002 | Quantitative descriptive | 1997-2000 | Work-related hearing loss | UK | All |
| 28 | Suuronen K [55] | 2007 | Quantitative descriptive | 1992-2001 | Occupational dermatitis and allergic respiratory diseases | Finland | Machine-related industries |
| 29 | Leigh J [56] | 2002 | Quantitative descriptive | 1945-2000 | Mesothelioma | Australia | All |
| 30 | Vainauskas S [57] | 2010 | Quantitative descriptive | 1999-2008 | Occupational Diseases | Lithuania | All |
| 31 | Turner S [58] | 2007 | Quantitative descriptive | 2002-2005 | Occupational skin diseases | UK | All |
| 32 | Szeszenia-Dąbrowska N [59] | 2013 | Quantitative descriptive | 1998-2011 | Occupational diseases | Poland | All |
| 33 | Szeszenia-Dąbrowska N [60] | 2006 | Quantitative descriptive | 2005 | Occupational diseases | Poland | All |
| 34 | Gobba F [61] | 2019 | Quantitative descriptive | 2012-2017 | non-melanoma skin cancers (NMSC) and actinic keratoses (AK) | Italy | Agriculture, Industry and Service sectors |
| 35 | Skov T [62] | 1990 | Quantitative descriptive | 1983-1987 | Occupational cancer | Denmark | All |
| 36 | Paris C [63] | 2012 | Quantitative descriptive | 2001-2009 | Work-related asthma | France | All |
| 37 | Barber CM [64] | 2019 | Quantitative descriptive | 1996-2015 | Occupational hypersensitivity pneumonitis (OHP) | UK | All |
| 38 | Hnizdo E [65] | 2001 | Quantitative descriptive | 1996-1998 | Occupational respiratory diseases | South Africa | Non-mining sector |
| 39 | Schmitt J [66] | 2014 | Quantitative descriptive | 2005-2011 | Occupational skin cancer due to UV radiation | Germany | All |
| 40 | Danø H [67] | 1996 | Quantitative descriptive | 1983-1990 | Occupational cancers (pleural mesothelioma, sinonasal adenocarcinoma, adenocarcinoma of the lung) | Denmark | All |
| 41 | Ameille J [68] | 2003 | Quantitative descriptive | 1996-1999 | Occupational asthma | France | All |
| 42 | Scarselli A [69] | 2010 | Quantitative descriptive | 1995-2008 | Occupational cancers | Italy | All |
| 43 | Nowak-Pasternak J [70] | 2022 | Quantitative descriptive | 2000-2019 | Silicoses | Poland | All |
| 44 | Carøe TK [71] | 2013 | Quantitative descriptive | 2000-2009 | Occupational skin cancer | Denmark | All |
| 45 | van der Molen HF [72] | 2020 | Quantitative descriptive | 2004-2017 | Occupational diseases | Italy | Agriculture |
| 46 | Urban M [73] | 2022 | Quantitative descriptive | 1991-2020 | Asbestoses | Czech Republic | All |
| 47 | Li X [74] | 2022 | Quantitative descriptive | 2006-2020 | Occupational cancer | China | All |

*(Continued)*

| No | Author | Year | Category of study design | Time period of data collection | Type of illness | Country | Industrial sector |
|---|---|---|---|---|---|---|---|
| 48 | Cherry NM [75] | 2001 | Quantitative descriptive | 1997-2000 | Work-related musculoskeletal diseases | UK | All |
| 49 | Swiątkowska B [76] | 2017 | Quantitative descriptive | 1970-2015 | Asbestos-related diseases (asbestosis, lung cancer, mesothelioma) | Poland | All |
| 50 | Kee D [77] | 2023 | Quantitative descriptive | 1996-2020 | Work-related musculoskeletal disorders | South Korea | All |
| 51 | Leigh J [78] | 1991 | Quantitative descriptive | 1982-1988 | Malignant mesothelioma | Australia | All |
| 52 | van Kampen V [79] | 2008 | Quantitative descriptive | 1970-2005 | Occupational respiratory diseases | Germany | All |
| 53 | Fenclova Z [80] | 2009 | Quantitative descriptive | 1992-2005 | Occupational hypersensitivity pneumonitis | Czech Republic | All |
| 54 | Jung S [81] | 2012 | Quantitative descriptive | 2001-2010 | Malignant mesothelioma | South Korea | All |
| 55 | Szeszenia-Dąbrowski N [82] | 2016 | Quantitative descriptive | 2000-2014 | Occupational disease | Poland | Agriculture |
| 56 | Cherry N [83] | 2000 | Quantitative descriptive | 1993-1999 | Occupational skin disease | UK | All |
| 57 | Vandenplas O [84] | 2011 | Quantitative descriptive | 1993-2002 | Occupational asthma | Belgium | All |
| 58 | Fagan KM [85] | 2016 | Qualitative | 2011 | Occupational injuries and illnesses | US | Poultry industry |
| 59 | Kwon S [86] | 2015 | Quantitative descriptive | 2004-2009 | Work-related asthma | South Korea | All |
| 60 | Parhar A [13] | 2011 | Quantitative non-RCT | N/A | Occupational asthma | Canada | All |
| 61 | Meyer JD [87] | 2001 | Quantitative descriptive | 1999 | Work-related respiratory diseases | UK | All |
| 62 | Cherry NM [88] | 2000 | Quantitative descriptive | 1996-1999 | Work-related diseases | UK | All |
| 63 | Kim KH [89] | 2010 | Quantitative descriptive | 1996-2009 | Work-related musculoskeletal disorders | South Korea | All |
| 64 | Money A [90] | 2011 | Quantitative descriptive | 1998-2006 | Work-related audiological disease | UK | All |
| 65 | Karjalainen A [91] | 2000 | Quantitative descriptive | 1989-1995 | Occupational asthma | Finland | All |
| 66 | Dulon M [92] | 2011 | Quantitative descriptive | 1998-2007 | Occupational airway diseases | German | Hair craft and health service |
| 67 | Aalto-Korte K [49] | 2019 | Quantitative descriptive | 2005-2016 | Occupational skin diseases | Finland | All |
| 68 | Alfonso JH [93] | 2015 | Quantitative descriptive | 2000-2013 | Work-related skin diseases | Norway | All |
| 69 | Oksa P [94] | 2019 | Quantitative descriptive | 1975-2013 | Occupational diseases | Finland | All |
| 70 | Carder M [95] | 2009 | Quantitative descriptive | 2002-2005 | Work-related mental ill health | UK | All |
| 71 | Meyer JD [96] | 2000 | Quantitative descriptive | 1996-1999 | Occupational contact dermatitis | UK | All |

*(Continued)*

| No | Author | Year | Category of study design | Time period of data collection | Type of illness | Country | Industrial sector |
|---|---|---|---|---|---|---|---|
| 72 | Latza U [97] | 2005 | Quantitative descriptive | 2003 | Occupational obstructive airway diseases | German | Industrial sector |
| 73 | Hussey L [98] | 2013 | Quantitative descriptive | 2006-2009 | Work-related ill-health | Great Britain | All |
| 74 | Zhou AY [99] | 2017 | Quantitative descriptive | 2001-2014 | Work-related ill-health and work-related mental ill health | Great Britain | Healthcare |
| 75 | Arnaud S [100] | 2010 | Quantitative non-RCT | November 2006 – February 2007 | Occupational diseases | France | N/A |
| 76 | Shum KW [101] | 2003 | Quantitative descriptive | 1993-1999 | Occupational contact dermatitis | UK | All |
| 77 | Halioua B [102] | 2012 | Quantitative descriptive | 2004-2007 | Occupational contact dermatitis | France | All |
| 78 | Carder M [103] | 2013 | Quantitative descriptive | 1996-2009 | Work-related mental ill-health and musculoskeletal disorders | UK | All |
| 79 | Shin S [104] | 2022 | Quantitative descriptive | 2001-2020 | Occupational infectious disease | South Korea | All |
| 80 | Cheng Y [105] | 2022 | Qualitative | July 2014 to March 2017 | Asbestos-related diseases | Taiwan | Shipbreaking, ship repairing, asbestos spraying in construction, and manufacturing and processing asbestos-containing products. |
| 81 | Hussey L [106] | 2008 | Quantitative descriptive | 2006-2007 | Work-related ill health | UK | All |
| 82 | Shih P [107] | 2023 | Quantitative descriptive | 2008-2021 | Occupational diseases | Taiwan | All |
| 83 | Malsam R [108] | 2021 | Quantitative descriptive | 2006-2019 | Occupational infectious diseases | German | Healthcare industries |
| 84 | Downs JW [109] | 2021 | Quantitative descriptive | 2008-2018 | Occupational poisonings | US | All |
| 85 | Carder M [110] | 2017 | Quantitative descriptive | 1996-2014 | Work-related, long-latency respiratory diseases | Great Britain | All |
| 86 | Cha E-W [111] | 2022 | Quantitative descriptive | 2020 | Work-related musculoskeletal disorders | South Korea | All |
| 87 | Carder M [112] | 2014 | Quantitative descriptive | 2005 & 2008 | Occupational disease | Great Britain | All |
| 88 | Carder M [113] | 2019 | Quantitative descriptive | 1989-2017 | Occupational and work-related respiratory disease attributed to cleaning products | UK | All |
| 89 | Hussey L [114] | 2010 | Quantitative descriptive | 2006-2007 | Work-related ill health | UK | All |
| 90 | Kourouklis GN [115] | 2009 | Quantitative descriptive | 2003-2007 | Occupational diseases | Greece | All |
| 91 | Barber CM [64] | 2018 | Quantitative descriptive | 1996-2017 | Silicosis | UK | All |
| 92 | Chen Y [116] | 2005 | Quantitative descriptive | 2004 | Occupational diseases | UK | Healthcare industries |
| 93 | Miedema HS [117] | 2013 | Quantitative descriptive | 2004-2011 | Low back pain related occupational diseases | Netherlands | All |

*(Continued)*

| No | Author | Year | Category of study design | Time period of data collection | Type of illness | Country | Industrial sector |
|---|---|---|---|---|---|---|---|
| 94 | Chen Y [118] | 2005 | Quantitative descriptive | 2002-2003 | Work-related musculoskeletal disorders | Great Britain | All |
| 95 | Stocks SJ [119] | 2015 | Quantitative descriptive | 2000-2012 | occupational asthma, contact dermatitis, noise-induced hearing loss, carpal tunnel syndrome and upper limb musculoskeletal disorders | 10 European countries: Belgium, the Czech Republic, Finland, France, Italy, the Netherlands, Norway, Spain, Switzerland and the UK | All |
| 96 | Moreno-Torres L [120] | 2018 | Quantitative descriptive | 2000-2015 | Occupational illnesses | Mexico | All |
| 97 | Alaguney ME [121] | 2020 | Quantitative descriptive | 2015 | Occupational diseases | Turkey | All |
| 98 | Scarselli A [122] | 2009 | Quantitative descriptive | 1994-2006 | Occupational cancer | Italy | Industrial sector |
| 99 | Grignoux J [123] | 2019 | Quantitative descriptive | 2001-2016 | Work-related laryngeal cancer | France | All |
| 100 | Garnett J [124] | 2020 | Quantitative descriptive | 2012-2019 | Occupational tuberculosis | South Africa | Healthcare industries |
| 101 | Kuijer P [125] | 2014 | Quantitative descriptive | 2005-2012 | Non-specific low back pain | Netherlands | All |
| 102 | Stocks SJ [126] | 2011 | Quantitative descriptive | 2002-2008 | Work-related ill health | UK | Construction industry |
| 103 | Carøe TK [71] | 2013 | Quantitative descriptive | 2010 | Occupational contact dermatitis | Denmark | All |
| 104 | Kim E [127] | 2021 | Quantitative descriptive | 1995-2017 | Malignant mesothelioma | South Korea | All |
| 105 | Money A [128] | 2015 | Quantitative descriptive | 2005-2012 | Work-related ill health | Republic of Ireland, Northern Ireland, Great Britain | All |
| 106 | Luckhaupt S [129] | 2010 | Quantitative descriptive | 1988 | Work-related chronic conditions | US | All |
| 107 | Chen Y [130] | 2005 | Quantitative descriptive | 2002-2003 | Work-related ill health | Scotland | All |
| 108 | Turner S [131] | 2015 | Quantitative descriptive | 1996-2012 | Work-related skin neoplasia | UK | All |
| 109 | Karttunen J [132] | 2013 | Quantitative descriptive | 1982-2008 | Occupational diseases | Finland | Agriculture |
| 110 | Plombom G [133] | 2015 | Quantitative descriptive | 2007-2012 | Occupational dermatitis | Brazil | All |
| 111 | Morken T [134] | 2007 | Quantitative descriptive | 1992-2003 | Work-related musculoskeletal disorders | Norway | Offshore petroleum industry |
| 112 | Stocks SJ [135] | 2010 | Quantitative descriptive | 2002-2008 | Work-related ill health | UK | Agriculture |
| 113 | Bensefa-Colas L [136] | 2015 | Quantitative descriptive | 2001-2010 | Occupational contact urticaria | France | All |

*(Continued)*

**Table 2.** (Continued)

| No | Author | Year | Category of study design | Time period of data collection | Type of illness | Country | Industrial sector |
|----|--------|------|--------------------------|-------------------------------|-----------------|---------|-------------------|
| 114 | Kersten JF [137] | 2020 | Quantitative descriptive | 2002-2017 | Occupational tuberculosis | German | Healthcare industries |
| 115 | Binazzi A [138] | 2021 | Quantitative descriptive | 1993-2018 | Malignant mesothelioma | Italy | Construction industry |
| 116 | Zhou AY [139] | 2017 | Quantitative descriptive | 2006-2009 | Work-related mental ill-health | Great Britain | Healthcare industries |
| 117 | Bensefa-Colas L [140] | 2014 | Quantitative descriptive | 2001-2010 | Occupational allergic contact dermatitis | France | All |
| 118 | Lysdal SH [141] | 2011 | Quantitative descriptive | 2009 | Hand eczema | Denmark | Service industry |
| 119 | McNamee R [142] | 2007 | Quantitative descriptive | 1996-2005 | Work-related skin and respiratory diseases | UK | All |
| 120 | Nienhaus A [143] | 2012 | Quantitative descriptive | 2005-2009 | Infectious diseases | German | Healthcare industries |
| 121 | Kanerva L [144] | 2000 | Quantitative descriptive | 1991-1997 | Occupational allergic contact dermatitis | Finland | All |
| 122 | Medeni I [145] | 2024 | Quantitative descriptive | 2018-2022 | Occupational diseases | Turkey | All |
| 123 | Fishwick D [146] | 2023 | Quantitative descriptive | 1998-2018 | Irritant asthma | UK | All |
| 124 | Karabağ I [147] | 2023 | Qualitative | 2021 | Occupational diseases | Turkey | All |
| 125 | Samant Y [148] | 2023 | Quantitative descriptive | 2022−2022 | COVID-19 | Norway | All |
| 126 | Su TY [149] | 2023 | Quantitative descriptive | 2004-2020 | Silicosis | Taiwan | All |
| 127 | Iskandar IYK [150] | 2024 | Quantitative descriptive | 1996-2019 | Occupational Diseases | UK | All |

### Reporting by type of disease

Occupational-related skin disease was the most studied health condition in Europe (18 studies), followed by cancer (16 studies), respiratory diseases (13 studies), musculoskeletal disorders (9 studies), asthma (9 studies), infectious diseases (8 studies), asbestoses (6 studies), mental illnesses (4 studies), hearing damage/loss (2 studies), poisoning (1 study), and chronic diseases (1 study). Other studies reported combined occupational diseases. Common occupational diseases reported by countries in the European region (UK, Poland, and Finland) included allergic contact dermatitis, irritant contact dermatitis, and contact urticaria as occupational diseases. However, in other regions (Asia, Africa, and Australia), skin diseases were less likely to be studied compared with other types of occupational diseases including musculoskeletal disorders, infectious diseases, and cancer.

Occupational cancer was the second top disease commonly reported in Europe and Asia. The following diagnoses were mostly reported in Europe: skin cancer (4 studies), mesothelioma (4 studies), laryngeal cancer (1 study), and all occupational cancers (2 studies). Many occupational cancers reported in Asia were mesothelioma (South Korean studies), followed by leukemia, and lung cancer (Chinese study).

The average number of reported cases of musculoskeletal disorders was highest in Europe and Asia. Lower back pain was the most frequently reported diagnosis across all musculoskeletal disorders in these two regions. Infectious diseases were also reported in Europe, Asia, and Africa, and were commonly found among healthcare workers. The types

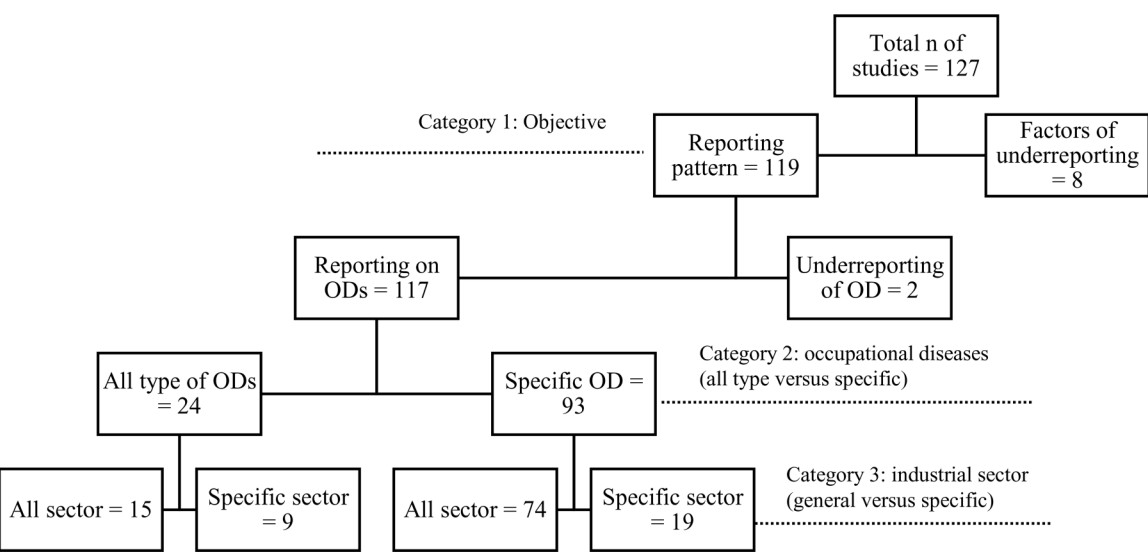

**Fig 2. Number of studies in each category based on study objectives, type of occupational diseases and industrial sector.**

of infectious diseases reported among the general working population were diarrheal, scabies, and tuberculosis, while hepatitis B and hepatitis C were frequently found in healthcare workers. Table 4 presents the most reported occupational diseases in the general working population classified by region. We included only studies that reported the number of cases of occupational disease in the general working population.

### Reporting of occupational diseases by industrial sector

In our findings, most reports were from health sector, agricultural, construction, mining and quarrying. Studies in the health sector were dominated by infectious diseases (n = 7), particularly tuberculosis, hepatitis, and scabies. Three of the seven studies on infectious diseases among healthcare workers were conducted in Germany, one in South Korea, one in South Africa, one in the Czech Republic, and one in Great Britain. One study reported on work-related mental illnesses among healthcare workers. Nurses had a higher number of cases of work-related mental illness compared to doctors. The only study that reported skin disease in healthcare workers was conducted in the Czech Republic. Allergic contact dermatitis, irritant contact dermatitis, and contact urticaria were the most frequently reported cases in the healthcare sector.

In the agricultural sector, we found only two studies: one analysed occupational dermatoses, and the other discussed non-melanoma skin cancers and actinic keratoses. The construction sector had only one study that reported on occupational cancer (i.e., malignant mesothelioma). Musculoskeletal disorders and hearing damage were reported among workers in the mining sector, while skin disease, asthma, and respiratory diseases were reported in the manufacturing sector. Table 5 presents commonly reported occupational diseases across the different industries.

### Underreporting of occupational diseases

Only two studies investigated the underreporting of occupational diseases at the country level. Both studies estimated a high underreporting rate of between 50% to 95%. One of the two studies (Moreno-Torres 2018) reported the underreporting of all types of occupational diseases, while Skov (1990) estimated the underreporting of occupational-related cancer among the general working population (Table 6).

**Table 3. Number of cases and incidence rate of occupational diseases.**

| Author, year | Reported period (year) | Country | Annual average n of cases (min-max) | IR per 100,000 employees |
|---|---|---|---|---|
| *All type of occupational diseases in all sectors* | | | | |
| *Data type: physician's report* | | | | |
| Cherry, 2000 [83] | 1996-1999 | UK | 10,941 | 42 |
| Chen, 2005(a) [130] | 2002-2003 | Scotland | 2,022 | 86 |
| Chen, 2005(b) [118] | 2002-2003 | UK, except Scotland | 21,120 | 84 |
| Hussey, 2008 [106] | 2006-2007 | UK | 1,436 | NA |
| Hussey, 2010(a) [114] | 2006-2007 | UK | 1,436 | NA |
| Hussey, 2010(b) [114] | 2006-2007 | UK | 1,680 | NA |
| van der Molen, 2012 [47] | 2009-2013 | Netherlands | 1,782 | 346 (330-362) |
| Hussey, 2013(a) [98] | 2006-2009 | UK | 1,171 | 1,387 |
| Hussey, 2013(b) [98] | 2006-2009 | UK | 7,759 | 131 |
| Carder, 2014 [112] | 2005-2007 | GB | 8,349 | 301 |
| | 2008-2010 | GB | 5,566 | 336 |
| Money, 2015 [128] | 2007-2012 | GB | 4,901 | NA |
| | | NI | 557 | NA |
| | | ROI | 161 | NA |
| Medeni, 2024 [145] | 2018-2022 | Turkey | 4,506 (3,231−5,952) | 15.78 (11.14-21.20) |
| *Data type: National Registry* | | | | |
| Shih, 2023 [107] | 2008-2021 | Taiwan | 2,038 (1,233−2,791) | NA |
| Szeszenia-Dąbrowska, 2006 [60] | 2005 | Poland | 3,249 | 35 |
| Vainauskas, 2010 [57] | 1999-2008 | Lithuania | 928 (570−1,447) | 64 (41-97) |
| Szeszenia-Dąbrowska, 2013 [59] | 1998-2011 | Poland | 5,035 (2,562−12,017) | 50 (25-117) |
| Oksa, 2019 [94] | 1975, 2005, 2013 | Finland | NA | 253 (200-310)* |
| *Data type: workers' compensation claims* | | | | |
| Kraut, 1994 [29] | 1989 | Canada | 37,927 | 28.2 |
| Shih, 2023 [107] | 2008-2021 | Taiwan | 752 (426−1,149) | NA |
| Kourouklis, 2009 [115] | 2003-2007 | Greece | 34 | 1.71 |
| Medeni, 2024 [145] | 2018-2022 | Turkey | 1,042 (955−1,209) | 3.64 (3.10-4.20) |
| *All type of occupational diseases in a specific sector* | | | | |
| *Agriculture sector* | | | | |
| Stocks, 2010 [135] | 2002-2008 | UK | 33 (19-46) | NA |
| Karttunen, 2013 [132] | 1982-2008 | Finland | 352 | NA |
| Szeszenia-Dąbrowska, 2016 [82] | 2000-2014 | Poland | 229 (141-340) | 12.8 (5-14.6) |
| van der Molen, 2020 [72] | 2004-2017 | Italy | 6431 | 1295 |
| Samant, 2020 [45] | 2007-2016 | Norway | 47 (26-66) | 114 (68-169) |
| *Health sector* | | | | |
| Walsh, 2005 [43] | 2002-2003 | UK | 5508 | NA |
| Zhou, 2017 [99] | 2001-2014 | GB | 146 | 515 |
| *Construction sector* | | | | |
| Stocks, 2011 [126] | 2002-2008 | UK | 264 | NA |

Note: IR = incidence rate; *Incidence rate in the study was measured in average incidence rate per 10,000 employees, this number has been adjusted to incidence per 100,000 employees. UK = United Kingdom, GB = Great Britain; NI = Northern Ireland; RoI = Republic of Ireland; NA = not available. Hussey, 2010(a) = cases reported by occupational physicians; Hussey, 2010(b) = cases reported by general practitioners. Hussey, 2013(a) = cases reported by general practitioners; Hussey, 2013(b) = cases reported by clinical specialists.

**Table 4. Most studied occupational diseases by region.**

| Region | Type of occupational disease | n of studies | n of years covered | Average n of annual cases (min-max) |
|---|---|---|---|---|
| Europe | Skin diseases | 16 | 6.7 (1 –14) | 1,268 (290–2,095) |
| | Respiratory diseases | 12 | 7.7 (1-23) | 1,384 (598–3,217) |
| | Cancer | 11 | 14.2 (3-36) | 160 (4 - 754) |
| | Musculoskeletal disorders | 5 | 5.6 (2 –12) | 1,653 (124 – 4,686) |
| | Asthma | 6 | 8.8 (3-20) | 288 (53–545) |
| | Infectious diseases | 5 | 3.5 (3 –4) | 737 (72 –1,402) |
| | Mental illnesses | 4 | 6.2 (3 –14) | 2,788 (526 − 4,709) |
| | Hearing damage/loss | 2 | 6 (3 –9) | 350 (244–540) |
| | Asbestoses | 4 | 23 (20 –30) | 58 (6–103) |
| Asia | Musculoskeletal disorders | 3 | 13.3 (1-25) | 4,537 (534−9,925) |
| | Cancer | 3 | 16 (10 –23) | 47 (28 - 75) |
| | Infectious diseases | 1 | 20 | 145 |
| | Asthma | 2 | 9.7 (6 –15) | 30 (14-39) |
| | Poisonings | 2 | 6.3 (1.6-11) | 645 (60 –1,194) |
| | Asbestoses | 2 | 13.5 (10 –17) | 3,911 (86 –7,736) |
| | Skin diseases | 1 | 1.6 | 68 |
| Northern America | Skin diseases | 1 | 1 | 4,289 |
| | Poisonings | 1 | 11 | 4,932 |
| Latin America | Skin diseases | 1 | 6 | 505 |
| Africa | Asthma | 1 | 2 | 113 |
| | Respiratory diseases | 1 | 2 | 1,530 |
| | Tuberculosis | 1 | 9 | 10 |
| Australia | Cancer | 2 | 14 (7 –20) | 182 (81-284) |

Note: n of studies defined as number of studies reported that particular occupational disease in the region; n of years covered defined as the average number of years of data collection from all studies; average n of annual cases defined as the average number of cases reported per year, calculated by averaging the total number of cases divided by the number of years of each study. One study might investigate more than one occupational disease and appear more than once in this table. Studies that reported other outcomes, e.g., incident rate, change of incident rate, are not included in the table.

## Factors affecting the underreporting of occupational diseases

We reviewed seven studies that investigated the underreporting of occupational diseases. Among these studies, one involved in-depth interviews with employees, another was an audit study of employers, and the remaining five were structured surveys targeting of different respondents including physicians, workers, and government representatives. A common theme across all studies was a concern among workers about the possible negative consequences related to job security when reporting occupational diseases. Alaguney et al [121] specifically highlighted this concern among subcontracted workers and those employed without a legal contract. Additionally, limited knowledge of the causal relationship between workplace risks and diseases, as well as lack of awareness of the reporting systems, emerged as key factors influencing the underreporting of occupational diseases (Table 7).

## Discussion

This systematic review aimed to estimate the global reporting and underreporting of occupational diseases and to identify factors that influence underreporting. Fifteen studies reported occupational diseases ranging from 34 to 37,927 cases per year with an annual incidence rate of 1.71 to 1,387 per 100,000 employees. This wide variation is largely due to the

**Table 5. Most reported occupational diseases by industrial sector.**

| Industrial sector | Type of occupational disease | n of studies | n of years covered | Average n of annual cases (min-max) |
|---|---|---|---|---|
| Health | Infectious disease | 3 | 8.7 (5 –14) | 80 (8–187) |
| | Tuberculosis | 4 | 9 (5 –16) | 99 (12-291) |
| | Mental illnesses | 1 | 4 | 239 (136-375)* |
| | Skin disease | 1 | 13 | 42 |
| Agriculture | Skin disease | 1 | 8 | 13 |
| | Cancer | 1 | 6 | 34 |
| Construction | Skin disease | 1 | 10 | 28 |
| | Asthma | 1 | 1 | 1,031 |
| | Respiratory disease | 2 | 4 (1 –10) | 245 (4-414) |
| | Cancer | 2 | 20 (13 –26) | 571 (480-661) |
| Mining and quarrying | Musculoskeletal disorder | 1 | 12 | 123 |
| | Respiratory disease | 1 | 10 | 6,183 |
| | Poisoning | 1 | 10 | 295 |

Note: In this table, we included studies which focused on a specific industrial sector and reported the number of occupational disease cases. Some studies investigated more than one type of occupational disease. Average number of cases per year is estimated by the total number of cases per year divided by total number of studies. *The cases were reported by general practitioners, psychiatrists, and occupational physicians.

**Table 6. Underreporting of occupational diseases.**

| Author, year | Reported period (year) | Country | Type of disease | n cases reported | n of cases not reported | Underreporting rate (%) |
|---|---|---|---|---|---|---|
| Skov, 1990 [62] | 1983-1987 | Denmark | Cancer | 78 | 178 | 50 |
| Moreno-Torres, 2018 [120] | 2000-2015 | Mexico | All | NA | NA | 89 (82 –95) |

non-uniformity in data reporting format and mechanism. Our review found that the rate of underreporting of occupational diseases was between 50% and 95%. The main factors explaining the high rate of underreporting include job insecurity, low awareness and knowledge of occupational diseases and the reporting process, inability to diagnose the disease, and lack of better occupational disease reporting policies.

The review included 127 eligible studies, predominantly from high-income countries, with only two articles from upper-middle-income countries, and none from lower-middle- and low-income countries, illustrating the wide gap between high and low/middle-income countries in occupational health research. Europe contributed the highest number of publications in occupational health research; a trend that could be linked to the continent's high level of industrialization. Historically, the first Industrial Revolution began in the United Kingdom in the 18th century and later spread to other countries [152]. This rapid industrialization not only changed the way businesses operate but also raised issues related to occupational health and safety, prompting more industrialized countries to establish laws and regulations to improve working conditions [152].

Despite these advances in high- and upper-middle-income countries, occupational health and safety have remained a lower-priority subject in LMICs compared to other health issues such as infectious diseases, non-communicable diseases, and malnutrition. For the papers included in this review, we found the UK to be the most active in contributing to the global evidence on occupational diseases, whereas none of the LMICs included in this review reported the number of occupational disease cases. What was most striking was the limited number of publications from China. Being the world's top manufacturing hub with the largest working population [153,154], one would expect more articles on occupational diseases. However, we found only three publications from China. There is the possibility that data on occupational diseases

**Table 7.** Factors contributing to the underreporting of occupational diseases.

| Author, year | Country | Type of OD | Method | Type of respondents | Contributing factors |
|---|---|---|---|---|---|
| Arnaud, 2010 [100] | France | All | Telephone study | Physicians | Difficulties in diagnosing ODs<br>Lack of knowledge of the reporting system<br>Perceived difficulties in the process of making a claim<br>Concerns about employee job security |
| Parhar, 2011 [13] | Canada | Asthma | Postal survey | Pulmonologists | Lack of knowledge of the reporting system<br>Concerns about employee job security |
| Lysdal, 2011 [141] | Denmark | Hand eczema | Postal survey | Hairdressers | Unaware of workplace risks and disease causation<br>Perceived difficulties in the claiming process<br>Lack of knowledge of the reporting system |
| Moldovan, 2017 [38] | 16 Eastern European countries | All | Online survey | Official national representatives | Employer's perceived as loss of revenue<br>Concerns about employee's job security<br>Improper monitoring by the authorities |
| Fagan, 2017 [85] | US | All | Inspections | Employers | Lack of supervision and training of the onsite medical unit<br>No clear structure and policies governing the onsite medical unit |
| Alaguney, 2020 [151] | Turkey | All | Online survey | Physicians | Employer's perceived as loss of revenue<br>Concerns about employee's job security, in particular, those working as subcontractor and those without a legal contract |
| Cheng, 2022 [105] | Taiwan | Asbestos-related diseases | In-depth interviews | Employees | Unaware of workplace risks and disease causation<br>Perceived difficulties in the claiming process<br>Lack of knowledge of the reporting system<br>Concerns about employee's job security |
| Karabağ, 2023 [147] | Turkey | | Modified delphi study | Physicians | Compensation-oriented system<br>Fear stigmatization and loss of income of the employee<br>Lack of occupational disease surveillance<br>Insufficient knowledge and experience of physicians<br>Inability to diagnose occupational diseases and insufficient training during medical education<br>Lack of an institutional strategy of Ministry of Health regarding the employment of occupational medicine specialists<br>Lack of recognition of the occupational medicine specialists<br>Low number of occupational medicine specialists<br>Lack of knowledge of the reporting system |

are published as government or institution reports, which we could not access due to language barriers. A previous study identified only six studies on occupational health conducted in LMIC between 1928 and 2019 [155]. A key challenge for many LMIC, aside from the gradual rise in industrialization, concerns data quality, completeness, utilization, and limited supporting health information infrastructure [156,157]. Despite having information systems in place, data in many LMICs are frequently reported without accompanying policies or regulations that facilitate data sharing and communication between stakeholders [158]. Without a proper system for data collection and interoperability, understanding the extent of occupational disease problems in these countries is difficult.

The working population in LMICs is larger than that of high-income countries (HICs) [159]. The United Nations World Population Prospects predicts that by 2030, the largest increases in population will be in LMICs, while populations in HICs will remain relatively stagnant [160]. Most of the population in LMICs will be in the productive age group, in contrast to the aging population in HICs. This demographic trend coupled with the growing industrialization in some middle-income countries, particularly China, necessitates the establishment of national surveillance systems to prevent and monitor occupational diseases. These systems would require significant investment in funding and human resources [161]. They would also require strong political support to make occupational health and safety one of the priority health concerns in LMICs

[162]. To date, fewer LMICs compared to HICs, have ratified the 2002 Protocol to the Occupational Safety and Health Convention, 1981 (No. 155), which was developed by the International Labour Organization (ILO) [163].

Occupational disease data can be sourced from national registries, physician reports, employers, employees, and workers' compensation claims. Our findings revealed that many European studies were based on physician reports and/or national registries while in Asia the studies relied more on data from worker compensation claims, and less on physician reports. Reporting of occupational disease is often hindered by a lack of knowledge about workplace risks and their causal relationship with diseases. Many employees have a low awareness of occupational hazards and their health implications [19,164]. Utilizing information collected by physicians could serve as an effective surveillance system, as physicians are better equipped to recognize occupational diseases compared to employers or employees. As our results demonstrate, the UK has the most comprehensive surveillance system that utilizes reports from general practitioners, occupational physicians, and other medical specialists (rheumatologists, pneumologists, audiologists, and infectious disease specialists) [165]. Not all physicians are able to easily be diagnosed occupational diseases due to limited training in diagnosing these diseases, including taking a detailed occupational history or hazard anamnesis. In addition, they may not have the time to thoroughly investigate workplace hazards, resulting in a lower priority for recording occupational diseases [166,167]. Although there is still underreporting of occupational diseases in the UK, their surveillance approach could be a good model for LMICs. Countries can adopt the voluntary physician reporting system, thereby reducing the reliance on employers for reporting. Further research to test the feasibility of voluntary physician reporting in low- and middle-income countries would be necessary.

Workers face various hazardous risks at their workplace, including exposure to chemical substances, physical hazards, biological agents, ergonomic stressors, and psychosocial hazards. Example of physical hazards include noise, vibration, radiation, and extreme temperature. Prolonged exposure to loud noises can cause hearing loss. Ergonomic stressors, such as poor posture, repetitive movements, and forceful exertions, are also present in some workplaces. When exposed to these stressors over a long period of time, the risk of musculoskeletal disorders would increase. Additionally, psychosocial hazards from heavy workloads, job stress, and anxiety can elevate the risk for mental disorders. In the health sector, biological and psychosocial hazards are usually dominated. In our review, tuberculosis, hepatitis B, and hepatitis C are the most frequently studied diseases among healthcare workers. Previous studies indicate that healthcare workers are frequently exposed to biological agents, such as blood and body fluids contaminated with bacteria or virus or parasites [168–173]. This result aligns with the findings of other studies showing that infectious diseases and mental illnesses are the most frequently reported occupational diseases in the health sector [174]. In the agricultural sector, workers are exposed to physical, ergonomic, biological, and chemical hazards. They are at risk of developing musculoskeletal disorders, cancer, and dermatitis, as evident from our findings. Looking at types of diseases by region, non-communicable diseases were more commonly reported in the studies conducted in America and Europe while respiratory and infectious diseases were most frequently studied in the African region. These findings align with the regional epidemiology: in America and Europe, where more than 70% of the disease burden is non-communicable, whereas in Africa only 30% is non-communicable [175]. It is worth noting that the diseases frequently studied are not necessarily the most common health problems.

The process of reporting occupational diseases involves many stakeholders, i.e., employers, employees, healthcare providers, and policymakers. With many layers of administrative procedures, the reporting process might be exhaustive and frustrating for some employees [176]. Before a case is reported to the authorities, the diagnosis must be confirmed by a physician. To do this, physicians should take a detailed occupational history of the worker, conduct a thorough physical examination, perform a laboratory test, if necessary, and undertake a careful evaluation to identify the link between the disease and exposure in the workplace [177]. This is not a straightforward process as it requires evidence. Physicians may lack motivation to investigate occupational diseases because of the time constraints and complex administrative procedures [178].

Artificial intelligence (AI) has the potential to help physicians and employers in reporting occupational diseases [179]. AI could be applied in monitoring occupational hazards, where the information generated can be used by employers and providers to create an effective prevention program for employees [180]. Additionally, the occupational hazards information can be analysed to predict possible occupational diseases in the workplace and simplify the reporting process [180,181]. Although there are numerous opportunities to apply AI in occupational health, we must be aware of the ethical and data privacy issues. Thus, more research around AI in occupational health is needed for better application of these tools.

## Strengths and limitations

Data sources varied substantially across the studies, providing limited information on the extent to which the working population was covered and demonstrating considerable methodological heterogeneity (e.g., differences in reporting systems, types of occupational diseases, and characteristics of the populations studied). This variation restricted our ability to synthesize the data. Nonetheless, given that most of the studies were of high quality (85.9% scored above 80 out of 100), the findings of this review are considered robust and can be interpreted with a high degree of confidence. Future studies assessing the national burden of occupational diseases should employ triangulation of different data sources to enhance data accuracy and validity.

While the findings of this study provide valuable insights into the reporting and underreporting of occupational diseases, readers must interpret these results with caution as they are drawn largely from what was reported in scientific publications and may not include government reports, which could lead to potential reporting bias. Although our search included grey literature, occupational disease statistics are not always publicly available. Furthermore, we identified 21 non-English publications out of 422 (4.98%) articles during our search, but these were not included in the analysis. We believe that excluding these non-English publications would not have any significant impact on the conclusions [182,183]. While few studies have evaluated the reporting and underreporting trends of occupational diseases using country-level data, to the best of our knowledge, this is the first systematic review summarising the global evidence on the reporting of occupational diseases.

## Conclusions

The evidence indicates a significant gap between high-income countries and LMICs in occupational health research. Occupational diseases are primarily reported in highly developed countries with established national reporting systems. Despite having a larger share of the global working population, LMICs lack evidence on occupational diseases, making it difficult to assess the magnitude of the occupational disease problem. Reporting mechanisms vary substantially with different countries relying on different data sources including worker compensation claims, disease registries, and physician reports. Our findings encourage policymakers, particularly in LMICs, to establish health information infrastructures for occupational disease reporting that enable data sharing and interoperability between stakeholders, including employers, employees, and physicians.

Countries could follow the practical guide on establishing a national system for recording and notification of occupational diseases which developed by the International Labour Organization. By having a standardized format on the reporting of occupational diseases, cross country comparisons can be done, and each country could monitor the prevalence of occupational diseases. This review also identified many barriers in reporting occupational diseases, such as employers or employees having limited knowledge of occupational disease reporting, employees being afraid of job insecurity, and doctors having difficulties in diagnosing occupational diseases. Thus, strengthening occupational health services is crucial for reducing the incidence of occupational diseases and improving the overall wellbeing of the global workforce. Occupational health services should encompass the health promotion, screening, diagnosis, disease surveillance, and reporting. Lastly, training in occupational health for healthcare workers, employers, and employees, would help to equip them with the knowledge and skills necessary to create a healthy work environment.

## Supporting information

**S1 File. Search strategies for electronic databases.**
(DOCX)

**S2 Fig. PRISMA flow diagram of study selection.**
(DOCX)

**S3 Table. Summary characteristics of all studies.**
(DOCX)

**S4 Table. Characteristics of each included studies.**
(DOCX)

**S5 Fig. Number of studies in each category based on study objectives, type of occupational diseases and industrial sector.**
(DOCX)

**S6 Table. Number of cases and incidence rate of occupational disease.**
(DOCX)

**S7 Table. Most studied occupational diseases by region.**
(DOCX)

**S8 Table. Most reported occupational diseases by industrial sector.**
(DOCX)

**S9 Table. Underreporting of occupational diseases.**
(DOCX)

**S10 Table. Factors contributing to the underreporting of occupational diseases.**
(DOCX)

**S11 File. List of excluded articles.**
(DOCX)

**S12 File. Risk of bias assessment of quantitative descriptive studies using Mixed Method Appraisal Tool (MMAT) version 2018.**
(DOCX)

**S13 File. Risk of bias assessment of quantitative non-randomized studies using Mixed Method Appraisal Tool (MMAT) version 2018.**
(DOCX)

**S14 File. Risk of bias assessment of qualitative studies using Mixed Method Appraisal Tool (MMAT) version 2018.**
(DOCX)

**S15 File. PRISMA 2020 Checklist.**
(DOCX)

## Acknowledgments

The authors would like to acknowledge University of New South Wales (UNSW) library for providing methodological guidance and access to scientific journals.

## Author contributions

**Conceptualization:** Levina Chandra Khoe, Muchtaruddin Mansyur, Virginia Wiseman, Augustine Asante.

**Data curation:** Levina Chandra Khoe, Siti Rizny Fitriana Saldi, Marsen Isbayuputra, Augustine Asante.

**Formal analysis:** Levina Chandra Khoe, Siti Rizny Fitriana Saldi, Marsen Isbayuputra, Augustine Asante.

**Funding acquisition:** Levina Chandra Khoe.

**Investigation:** Levina Chandra Khoe, Siti Rizny Fitriana Saldi, Marsen Isbayuputra.

**Methodology:** Levina Chandra Khoe, Siti Rizny Fitriana Saldi, Marsen Isbayuputra, Muchtaruddin Mansyur, Virginia Wiseman, Augustine Asante.

**Project administration:** Levina Chandra Khoe.

**Supervision:** Muchtaruddin Mansyur, Virginia Wiseman, Augustine Asante.

**Validation:** Siti Rizny Fitriana Saldi, Marsen Isbayuputra, Muchtaruddin Mansyur, Virginia Wiseman, Augustine Asante.

**Writing – original draft:** Levina Chandra Khoe.

**Writing – review & editing:** Levina Chandra Khoe, Siti Rizny Fitriana Saldi, Marsen Isbayuputra, Muchtaruddin Mansyur, Virginia Wiseman, Augustine Asante.

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
