## [Decision Letter · Decision Letter 0]

10 Mar 2025

Dear Dr. Khoe,

Thank you for submitting your manuscript to PLOS ONE. After careful consideration, we feel that it has merit but does not fully meet PLOS ONE’s publication criteria as it currently stands. Therefore, we invite you to submit a revised version of the manuscript that addresses the points raised during the review process.

To improve the manuscript, please carefully consider the reviewers' comments and address each one thoroughly. Pay particular attention to the feedback from reviewers #1 and #5. Additionally, ensure that the definitions of occupational diseases are clearly established from the outset, as highlighted by the reviewers. Address objective number 2 explicitly in the conclusion and discussion sections. Clearly discuss the gaps observed in reporting from LMICs and propose methods to better capture LMIC data, such as alternative data sources or partnerships with local agencies.

Finally, please revise the manuscript carefully for grammatical and spelling errors throughout

We look forward to receiving your revised manuscript.

Kind regards,

Tesfaye Hambisa Mekonnen

Academic Editor

PLOS ONE

Journal Requirements:

2. Please ensure that your PRISMA flow diagram is included in your main manuscript file as Figure 1; please see the PLOS ONE submission guidelines for systematic reviews and meta-analyses at https://journals.plos.org/plosone/s/submission-guidelines#loc-systematic-reviews-and-meta-analyses .

“This research is part of L.C.K doctoral study, which was supported by the Indonesian Endowment Funds for Education (LPDP).”

5. We note that your Data Availability Statement is currently as follows: “All relevant data are within the manuscript and in Supporting Information files.”

7. Please upload a copy of Figure 1 and 2, to which you refer in your text on page 10 and 12. If the figure is no longer to be included as part of the submission please remove all reference to it within the text.

8. As required by our policy on Data Availability, please ensure your manuscript or supplementary information includes the following:

Additional Editor Comments:

Dear Dr Khoe

Thank you for submitting your manuscript to PLOS ONE. After careful consideration, we feel that it has merit but does not fully meet PLOS ONE’s publication criteria as it currently stands. Therefore, we invite you to submit a revised version of the manuscript that addresses the points raised during the review process.

To improve the manuscript, please carefully consider the reviewers' comments and address each one thoroughly. Pay particular attention to the feedback from reviewers #1 and #5. Additionally, ensure that the definitions of occupational diseases are clearly established from the outset, as highlighted by the reviewers. Address objective number 2 explicitly in the conclusion and discussion sections. Clearly discuss the gaps observed in reporting from LMICs and propose methods to better capture LMIC data, such as alternative data sources or partnerships with local agencies. Finally, please revise the manuscript carefully for grammatical and spelling errors throughout

Reviewers' comments:

Reviewer's Responses to Questions

**Comments to the Author**

1. Is the manuscript technically sound, and do the data support the conclusions?

Reviewer #1: Yes

Reviewer #2: Partly

Reviewer #3: Yes

Reviewer #4: Yes

Reviewer #5: Partly

2. Has the statistical analysis been performed appropriately and rigorously?

Reviewer #1: Yes

Reviewer #2: Yes

Reviewer #3: N/A

Reviewer #4: N/A

Reviewer #5: Yes

3. Have the authors made all data underlying the findings in their manuscript fully available?

Reviewer #1: Yes

Reviewer #2: Yes

Reviewer #3: Yes

Reviewer #4: Yes

Reviewer #5: Yes

4. Is the manuscript presented in an intelligible fashion and written in standard English?

Reviewer #1: Yes

Reviewer #2: Yes

Reviewer #3: Yes

Reviewer #4: Yes

Reviewer #5: No

Reviewer #1: The figures are called out in the main text but listed as S3 appendices--please either put them in the main document or refer to them in the appendix.

In your conclusions (abstract) you call for the need for a national system of reporting occupational diseases, do you mean international, so we have a better global count? Or are you meaning each country should establish a national system?

Background, line 78: Suggest changing wording to say "exposure typically is sourced" from employers. Other times, likelihood of exposure is sourced from a 3rd party (IH for example) especially when its occurring in a court of law

Background, line 112: you mention the review will bring together evidence on LMICs, but my understanding is no studies in low income countries were included

For eligibility criteria, did you include modeling studies?

Quality Assessment: more details are needed on the scoring between 20-100. Is there firm criteria or is it semi-quantitative? Given at least two reviewers applied the tool, do you have information on the agreement between the reviewers? Were scores averaged or was the rubric they applied so strict that they had to agree exactly?

Results: Characteristics of included studies may be able to be displayed in a table that number of studies by year, country/region, income, disease, industry sector, etc. This would not be the detail in S2, just a summary as presented in the text.

Results line 308: "Occupational-related skin disease was the most reported condition in Europe"--was it the most reported or was it the most studied? Or the condition which had the most included studies? Things like skin disease are easier to study for many reasons than cancers but doesn't mean they are the most common work-related condition. In general, I'd just be careful with this language to make it clear these were the most common studies in the review not necessarily the most common in the population

line 368: Make it clear these are only 2 studies in the review that looked at underreporting, as many other studies have investigated underreporting of occ diseases

I like the inclusion of the exploration of factors affecting underreporting of occupational diseases and am pleased to see that in the paper!

Line 395: is it 128 studies or 127?

Line 411: Was this really the most striking thing?

Line 456: Is TB, HepB and HepC the most common health concerns, or the most frequently studied? I would think MSDs and stress/mental health would also be extremely common health concerns although less frequently studied. The studies you cite look at the prevalence of these concerns, but this doesn't mean they are the most common, it just means they were studied and you found them in your review. I think it's important to acknowledge your systematic review is far from compelte and is looking at diseases that people decided to look at--making claims like "communicable are more likely in Africa" v. "Non-communicable is more likely in Europe" Is a function of what the literature has looked at--the entire scope of possible diseases has not been considered. Using language like, "in our review" or "in the papers included in this review" could help to temper some of the findings

Nowhere in the discussion is the risk of bias talked about and how this could impact your findings

There are lots of places that occupational disease are reported other than in the literature. For example, through ministries/departments of labor. Overall, I think conflating what is in the literature as what is reported is in error (example: line 482).

Reviewer #2: Hello

This is a good article and a good topic has been chosen.

In my opinion, the conclusion section should be more complete. Currently, only the conclusion has been drawn that the real conditions do not match the existing reports, which we could have predicted almost without conducting this study.

It would be better to provide better solutions based on the results of this study, especially suggestions for conducting future studies.

Good luck.

Reviewer #3: I carefully reviewed the manuscript. The aim of the manuscript is to give an overall profile of the world occupational diseases' trends via a systematic review. I found this manuscript useful as an acceptable report on the situation of the workforce health all around the world. We know that reporting systems even in High Income Countries are not working good enough and many people lost their health as a result of poor occupational health services in their countries. From this point of view, this manuscript has a great value for sensitize the researchers and policymakers to pay more attention to the subject.

Most of my concerns about the rigor of the study are cover and answered in Limitation section. However, authors can strengthen the Conclusion by means of the mapping of the findings on the elements of a good occupational health service including screening, registry, primary health care services, secondary, and third health care services to cover the workforce health. They also can better describe the reasons underlying factors for underreporting the OD cases.

As minor a minor comment, I saw the figure, 127, for the number of the studies considered in Discussion, Line 395. while in the other section, this number is reported 128!

Reviewer #4: Abstract:

In the abstract method section, mention the literature search period to increase transparency.

Introduction:

The opening sentence could be stronger by highlighting the direct impact of underreporting on health policies and worker protection.

Methods:

"This study is registered on PROSPERO, number CRD42023417814."

This sentence appears twice (lines 125 and 122). You can delete one of them to avoid repetitiveness.

Discussion:

Some sections discuss LMICs in depth, but comparisons with HICs are not always clear. Emphasis could be added on why LMICs lag behind in reporting and how HICs can be a model for improvement.

Reviewer #5: All comments upload as an attachment....................................................................................................................................................................

**Do you want your identity to be public for this peer review?** For information about this choice, including consent withdrawal, please see our Privacy Policy

Reviewer #1: No

Reviewer #2: No

Reviewer #3: **Yes:** Dr. Mostafa Pouyakian

Reviewer #4: **Yes:** Abdul Rohim Tualeka

Reviewer #5: No

---

## [Author Response · Author response to Decision Letter 1]

30 Apr 2025

Point-by-point response

Reviewer #1

The figures are called out in the main text but listed as S3 appendices--please either put them in the main document or refer to them in the appendix.

Response: Thank you, we have included the figures in the main text.

In your conclusions (abstract) you call for the need for a national system of reporting occupational diseases, do you mean international, so we have a better global count? Or are you meaning each country should establish a national system?

Response: Our study included articles that assessed the burden of occupational diseases at country level (line 141-142). We found no studies from low- and lower-middle-income countries, indicate the need for reporting systems in those countries. Therefore, in the conclusion (abstract), we encourage that every country should establish a national reporting system, which will contribute to the global statistics of occupational diseases.

Background, line 78: Suggest changing wording to say "exposure typically is sourced" from employers. Other times, likelihood of exposure is sourced from a 3rd party (IH for example) especially when its occurring in a court of law

Response: Thanks for your suggestion. We have changed the wording as suggested in the Background, line 89.

Background, line 112: you mention the review will bring together evidence on LMICs, but my understanding is no studies in low income countries were included

Response: Yes, originally our aim was to provide evidence on occupational disease reporting from countries based on their income levels. However, our study found that no studies have been conducted in low income countries. Thus, we changed the wording from “both high-income countries and LMICs” to “all countries” (Background, line 125).

For eligibility criteria, did you include modeling studies?

Response: We focused on the descriptive or explanatory studies that provide real world data at the national level, but not for predicting or forecasting outcomes. Thus, we excluded the modelling studies (added in Eligibility Criteria line 146).

Quality Assessment: more details are needed on the scoring between 20-100. Is there firm criteria or is it semi-quantitative? Given at least two reviewers applied the tool, do you have information on the agreement between the reviewers? Were scores averaged or was the rubric they applied so strict that they had to agree exactly?

Response: Five criteria were used to assess the quality of each study. Studies were scored as follows:

• 0% of quality criteria met if there is no “yes” answer to any of the five criteria;

• 20% of quality criteria met if there is one “yes” answer;

• 40% of quality criteria met if there are two “yes” answers;

• 60% for three “yes” answers;

• 80% for four “yes” answers”; and

• 100% for “yes” answers to all questions.

At the beginning of the appraisal process, two investigators (LK or SR / LK or MI) independently assessed the papers. If different results were obtained, the assigned investigators discussed until an agreement was reached. If no agreement was reached, a third investigator (SR or MI) was consulted to obtain an agreement. The statement about how to resolve any disagreement has been added in the Methods section line 212-214.

Results: Characteristics of included studies may be able to be displayed in a table that number of studies by year, country/region, income, disease, industry sector, etc. This would not be the detail in S2, just a summary as presented in the text.

Response: Thank you for your suggestion. We have incorporated table 1 in the manuscript to depict the characteristics of included studies.

Results line 308: "Occupational-related skin disease was the most reported condition in Europe"was it the most reported or was it the most studied? Or the condition which had the most included studies? Things like skin disease are easier to study for many reasons than cancers but doesn't mean they are the most common work-related condition. In general, I'd just be careful with this language to make it clear these were the most common studies in the review not necessarily the most common in the population

Response: Thanks for your suggestion. We agree with your suggestion and have changed the wording to “the most studied condition” (line 335).

line 368: Make it clear these are only 2 studies in the review that looked at underreporting, as many other studies have investigated underreporting of occ diseases

Response: In this study, we included only studies that assess the reporting or underreporting of occupational diseases at the country level (line 141-142). Thus, we excluded studies on reporting or underreporting of occupational diseases conducted at the facility or company level. We excluded these studies because we aims to have a summary of global trend of occupational diseases and make a comparison between countries.

I like the inclusion of the exploration of factors affecting underreporting of occupational diseases and am pleased to see that in the paper!

Response: Thank you!

Line 395: is it 128 studies or 127?

Response: Apologies for the mistake, it is 127 studies, as referred to in Figure 1.

Line 411: Was this really the most striking thing?

Response: As the world’s top manufacturer with the largest population, we assumed that China might have a significant burden of occupational diseases, which could be reflected in a higher number of publications on the topic. However, contrary to our expectations, there were only three publications from China, all in Chinese, and none in English. The limited number of publications does not necessarily mean that China has a low burden of occupational diseases; rather, it might indicate that limited research had been done in this area.

Line 456: Is TB, HepB and HepC the most common health concerns, or the most frequently studied? I would think MSDs and stress/mental health would also be extremely common health concerns although less frequently studied. The studies you cite look at the prevalence of these concerns, but this doesn't mean they are the most common, it just means they were studied and you found them in your review.

Response: Thank you for this important comment. We agreed that TB, HepB, and HepC are the most frequently studied and not necessarily the most common health concerns. We changed the sentence accordingly in line 503-504. Nevertheless, previous studies have indicated that healthcare workers are frequently exposed to biological agents, and therefore have a higher prevalence of TB and hepatitis than the general population. We have added additional references to support this argument (Tarantola et al 2006 & Reis et al 2019).

I think it's important to acknowledge your systematic review is far from complete and is looking at diseases that people decided to look at--making claims like "communicable are more likely in Africa" v. "Non-communicable is more likely in Europe" Is a function of what the literature has looked at--the entire scope of possible diseases has not been considered. Using language like, "in our review" or "in the papers included in this review" could help to temper some of the findings

Response: We changed the wording in line 508-516 to avoid misinterpretation of the findings, and used the terms “in our review”, and “frequently studied”.

Nowhere in the discussion is the risk of bias talked about and how this could impact your findings.

Response: We have added a statement in the discussion section (line 515-516) to caution readers to be careful when interpreting the results of this review.

There are lots of places that occupational disease are reported other than in the literature. For example, through ministries/departments of labor. Overall, I think conflating what is in the literature as what is reported is in error (example: line 482).

Response: We agreed that occupational diseases can be reported not only from literatures, but also from government or institution reports. Thus, we included WHO Institutional Repository for Information Sharing (IRIS), Dimensions, and Google Scholar in our searching databases, as described in the Methods section. The result showed that many papers were published from high-and upper-middle-income countries, and none from low- and lower-middle-income countries, indicating a significant gap in the occupational health research.

Reviewer #2:

Hello

This is a good article and a good topic has been chosen.

In my opinion, the conclusion section should be more complete. Currently, only the conclusion has been drawn that the real conditions do not match the existing reports, which we could have predicted almost without conducting this study.

It would be better to provide better solutions based on the results of this study, especially suggestions for conducting future studies.

Good luck.

Response: Thank you for your suggestion. We have added a recommendation to follow the practical guide for establishing a national system for recording and notification of occupational diseases, which was developed by the International Labour Organization (line 569-571).

Reviewer #3:

I carefully reviewed the manuscript. The aim of the manuscript is to give an overall profile of the world occupational diseases' trends via a systematic review. I found this manuscript useful as an acceptable report on the situation of the workforce health all around the world. We know that reporting systems even in High Income Countries are not working good enough and many people lost their health as a result of poor occupational health services in their countries. From this point of view, this manuscript has a great value for sensitize the researchers and policymakers to pay more attention to the subject.

Response: Thank you, we appreciate your thorough review.

Most of my concerns about the rigor of the study are cover and answered in Limitation section. However, authors can strengthen the Conclusion by means of the mapping of the findings on the elements of a good occupational health service including screening, registry, primary health care services, secondary, and third health care services to cover the workforce health. They also can better describe the reasons underlying factors for underreporting the OD cases.

Response: We added some statements regarding the reasons for underreporting and the importance of strengthening occupational health services (line 569-573).

As minor a minor comment, I saw the figure, 127, for the number of the studies considered in Discussion, Line 395. while in the other section, this number is reported 128!

Response: Thanks for your comments. This was a typographical error; the correct number of studies is 127. We have revised this accordingly.

Reviewer #4: Abstract:

In the abstract method section, mention the literature search period to increase transparency.

Response: We added the literature search period in the abstract.

Introduction:

The opening sentence could be stronger by highlighting the direct impact of underreporting on health policies and worker protection.

Response: Thank you for your suggestion. In the Introduction section, we have added line 68-70 to highlight more strongly the impact of occupational diseases on workers’ health. We have also added line 120-122 to highlight the consequences of underreporting, especially its negative impact on the effectiveness of prevention efforts.

Methods:

"This study is registered on PROSPERO, number CRD42023417814."

This sentence appears twice (lines 125 and 122). You can delete one of them to avoid repetitiveness.

Response: We have removed the repeated sentence (line 125 in the original manuscript).

Discussion:

Some sections discuss LMICs in depth, but comparisons with HICs are not always clear. Emphasis could be added on why LMICs lag behind in reporting and how HICs can be a model for improvement.

Response: Thank you for your comments. In the discussion section, we provided arguments regarding the challenges of reporting occupation-related diseases in LMICs (line 418-424). We also added a statement about occupational disease surveillance in the UK and mentioned that this approach could be a good example for LMICs to follow (line 488-491).

Reviewer #5:

All comments upload as an attachment.

1. Abstract

1.2. Keywords should be modified according to the title and topic of the research.

Response: The following keywords are used in this paper: occupational disease; underreporting; mandatory reporting; workers' compensation. We have added “global trends” to the list. We believe these keywords are relevant to the topic of the research, i.e., underreporting of occupational diseases.

2.2. Use the passive voice in writing the abstract and the entire article.

Response: Thank you for your suggestions. We have revised several sentences in the manuscript using the passive voice, where appropriate. However, we believe that both passive and active voices are important but would be open to further advice from the editors on where further changes should be made.

References:

1. Kojima T, Popiel HA. Proper Scholarly Writing for Non-Native English-Speaking Authors: Choosing Active and Passive Voice, Rewording, and Refining Texts. J Korean Med Sci. 2022 Nov 14;37(44):e312. doi: 10.3346/jkms.2022.37.e312. PMID: 36377291; PMCID: PMC9667014.

2. Millar N, Budgell BS. The passive voice and comprehensibility of biomedical texts: An experimental study with 2 cohorts of chiropractic students. J Chiropr Educ. 2019 Mar;33(1):16-20. doi: 10.7899/JCE-17-22. Epub 2018 Aug 2. PMID: 30070902; PMCID: PMC6417867.

2. Introduction

2.1 The introduction should provide a more comprehensive overview of the significance of reporting and underreporting of occupational disease, outlining its historical development and current relevance.

2.2. Include more recent references from the last two years to ensure the literature review is up-to-date.

2.3. It is best to explain the difference between occupational and non-occupational diseases, as well as the difficulties in distinguishing between the two. In addition, you are expected to give examples and explanations of common occupational diseases.

2.4. The studies conducted in the study focused mostly on World Health Organization statistics, and it seems that reputable research and articles have not been examined much.

Response: Thank you for your suggestions. We added references from recent publications to provide greater global context on the burden of occupational diseases in the Introduction section (line 60-61). Additionally, we have explained the difference between occupational and non-occupational diseases in the Introduction section (line 60-69). Our review included five electronic databases and three other sources (i.e., WHO IRIS, Dimensions, and Google Scholar) to identify grey literatures. World Health Organization statistics are presented in the Introduction section. We also included studies that used real world data from various countries, as described in the eligibility criteria. The search was updated in September 2024 to ensure all relevant studies are captured.

3. Methods

3.1. Include more visual aids, such as charts, graphs, and tables, to illustrate key points and data trends.

Response: The study selection process is illustrated in Figure 1, depicting the study flow and number of studies in PRISMA flowchart.

3.2. Provide a critical analysis of the reviewed studies, highlighting any methodological weaknesses or biases.

Response: All included studies were critically appraised using the Mixed Methods Appraisal Tool (Quality Assessment subsection in Methods, line 200-209).

4. Results

4.1. Only two studies on underreporting are mentioned in the study. These two studies alone cannot provide a proper analysis of the current situation and identify the factors influencing this.

Response: In our opinion, reporting and underreporting of occupational diseases are two sides of the same coin. The more we know about reporting, the better we understand the degree of underreporting. The two specific studies on underreporting mentioned in this review may not, on their own, provide sufficient insight into the current state of underreporting of occupational

---

## [Decision Letter · Decision Letter 1]

7 Oct 2025

Dear Dr. Khoe,

Thank you for submitting your manuscript to PLOS ONE. After careful consideration, we feel that it has merit but does not fully meet PLOS ONE’s publication criteria as it currently stands. Therefore, we invite you to submit a revised version of the manuscript that addresses the points raised during the review process.

We look forward to receiving your revised manuscript.

Kind regards,

Alejandro Torrado Pacheco, PhD

Staff Editor

PLOS ONE

Journal Requirements:

Additional Editor Comments:

The manuscript has been evaluated by three reviewers, and their comments are generally positive. However some minor points still require clarification. Please address the following:

1) Line 198: "the study was given a score of between 20 and 100". Is there firm criteria or is it semi-quantitative? Given at least two reviewers applied the tool, do you have information on the agreement between the reviewers? Were scores averaged or was the rubric they applied so strict that they had to agree exactly? Please add further details of the risk of bias assessment to clarify.

2) Further discussion is needed around the risk of bias and how this could impact your findings.

3) Abstract: line 26 "the work activity" is clunky, should be rephrased, e.g. "work activities". In general please proofread for typographical and grammatical errors

4) please move table S2 into the main text

5) please provide reference details for all of the papers mentioned in tables S2, S4, S6, S7, S8 etc. Essentially, wherever a paper is mentioned there should be a reference. If the references of the 129 papers are included in the main reference list then they can be referred to by number, i.e. Ding, 2013 [67] etc.

Reviewers' comments:

Reviewer's Responses to Questions

**Comments to the Author**

Reviewer #2: All comments have been addressed

Reviewer #3: All comments have been addressed

Reviewer #5: All comments have been addressed

2. Is the manuscript technically sound, and do the data support the conclusions?

Reviewer #2: Partly

Reviewer #3: (No Response)

Reviewer #5: Yes

3. Has the statistical analysis been performed appropriately and rigorously?

Reviewer #2: I Don't Know

Reviewer #3: (No Response)

Reviewer #5: Yes

4. Have the authors made all data underlying the findings in their manuscript fully available?

Reviewer #2: Yes

Reviewer #3: (No Response)

Reviewer #5: Yes

5. Is the manuscript presented in an intelligible fashion and written in standard English?

Reviewer #2: Yes

Reviewer #3: (No Response)

Reviewer #5: Yes

Reviewer #2: Hello

Thanks to the authors of the article

Quite good explanations have been added and the article has become better

Good luck

Reviewer #3: (No Response)

Reviewer #5: (No Response)

**Do you want your identity to be public for this peer review?** For information about this choice, including consent withdrawal, please see our Privacy Policy

Reviewer #2: No

Reviewer #3: **Yes:** Dr. Mostafa Pouyakian

Reviewer #5: **Yes:** ali alboghobeish

---

## [Author Response · Author response to Decision Letter 2]

19 Nov 2025

Dear Editor,

We sincerely thank the editor and reviewers for taking the time and effort to review our revised manuscript entitled “Global Trends in Occupational Disease Reporting: A Systematic Review” (PONE-D-24-50153R1). We have carefully reviewed the comments and revised the manuscript accordingly.

Kindly find our responses to editor comments in a point-by-point manner below. Any changes are highlighted within the manuscript. We hope the revised version is now suitable for publication in PLOS One Journal and look forward to hearing from you in due course.

Yours sincerely,

On behalf of all authors,

Levina Chandra Khoe

Additional Editor Comments:

The manuscript has been evaluated by three reviewers, and their comments are generally positive. However some minor points still require clarification. Please address the following:

1) Line 198: "the study was given a score of between 20 and 100". Is there firm criteria or is it semi-quantitative? Given at least two reviewers applied the tool, do you have information on the agreement between the reviewers? Were scores averaged or was the rubric they applied so strict that they had to agree exactly? Please add further details of the risk of bias assessment to clarify.

Response:

Thank you for your comment. If there is a difference in scores, two reviewers will discuss it to reach a mutual agreement. If no agreement is reached, they will involve a third reviewer to discuss the matter further.

In the previously revised manuscript (file name: Manuscript SR OD Report_clean_300425), the following information was added (line 216-218):

“If different results were obtained, the assigned investigators discussed until an agreement was reached. If no agreement was reached, a third investigator (SR or MI) was consulted to obtain an agreement.”

2) Further discussion is needed around the risk of bias and how this could impact your findings.

Response:

We have added the following to the Discussion section to address the potential risk of reporting bias and language bias (line 550-557):

“While the findings of this study provide valuable insights into the reporting and underreporting of occupational diseases, readers must interpret these results with caution. The results are drawn largely from what was reported in scientific publications and may not include government reports, which could lead to potential reporting bias. Although our search included grey literature, occupational disease statistics are not always publicly available. Furthermore, we identified 21non-English articles out of 422 (4.98%) during our search, but these were not included in the analysis. We believe that excluding these non-English publications would not have any significant impact on the conclusions (59, 60).”

3) Abstract: line 26 "the work activity" is clunky, should be rephrased, e.g. "work activities". In general please proofread for typographical and grammatical errors

Response:

Thank you, we have revised accordingly.

4) please move table S2 into the main text

Response:

Thank you, we have moved table S2 into the main text as suggested and renamed it as Table 1.

5) please provide reference details for all of the papers mentioned in tables S2, S4, S6, S7, S8 etc. Essentially, wherever a paper is mentioned there should be a reference. If the references of the 129 papers are included in the main reference list then they can be referred to by number, i.e. Ding, 2013 [67] etc.

Response:

Thank you, we have provided the reference details as requested.

---

## [Editor Report · Decision Letter 2]

18 Dec 2025

Dear Dr. Khoe,

 Thank you for your latest revisions. We still feel that some of the points raised in the last decision require clarification. Specifically: 1. Line 198: "the study was given a score of between 20 and 100". Can you please clarify in detail how these scores were assigned? Did every "yes" answer equate to a score of 20, with a maximum score of 100 for five "yes" answers? Please also clarify what those scores represent in terms of assessed quality, and how they were then used in the systematic review. For example, were any studies excluded based on the quality assessment (such as the study that received a score of 0/100)? 2. For the discussion of risk of bias, we mean this in the sense of the risk of bias assessment (also known as quality assessment) that was performed using the MMAT tool. Please discuss how this could impact your findings.  3. Lastly, please change the title of your manuscript, as the word "trends" does not correspond to the content, as you did not assess trends over time. An appropriate title could be: "Global reporting and under-reporting of occupational diseases: a systematic review".

We look forward to receiving your revised manuscript.

Kind regards,

Alejandro Torrado Pacheco, PhD

Staff Editor

PLOS One
---

## [Author Response · Author response to Decision Letter 3]

26 Jan 2026

Comments from editor:

Thank you for your latest revisions. We still feel that some of the points raised in the last decision require clarification. Specifically:

Line 198: "the study was given a score of between 20 and 100". Can you please clarify in detail how these scores were assigned?

There are five criteria in each study design category. A “yes” response indicates that the paper provides sufficient information for that criterion. A “no” response indicates that the paper does not report any information relevant to the criterion, while a “can’t tell” response indicates that the available information is unclear or insufficient to determine whether the criterion is met.

Each “yes” answer is assigned a score of 20. A score of 0 is assigned for every “no” or “can’t tell” response.

Did every "yes" answer equate to a score of 20, with a maximum score of 100 for five "yes" answers?

Yes, every “yes” answer equates to a score of 20, with a maximum score of 100 for five “yes” answers.

Please also clarify what those scores represent in terms of assessed quality, and how they were then used in the systematic review. For example, were any studies excluded based on the quality assessment (such as the study that received a score of 0/100)?

A maximum score of 100 indicates that the paper reported all the necessary information for each criterion. Conversely, a score of zero means the paper did not contain any information relevant to any of the criteria. Studies with lower scores are therefore considered to have greater methodological limitations and a higher risk of bias. No studies were excluded from this review based on the quality assessment, as all had already met the inclusion criteria during the screening stage. Thus, the primary purpose of the quality assessment was to determine the trustworthiness and scientific soundness of the included studies. This exercise enabled us to judge the extent to which the findings of each study might have been influenced by poor design or methodological flaws. A significant proportion of studies in this systematic review (over 80%) achieved high quality scores.

2. For the discussion of risk of bias, we mean this in the sense of the risk of bias assessment (also known as quality assessment) that was performed using the MMAT tool. Please discuss how this could impact your findings.

The risk of bias assessment indicates a high level of methodological quality across the included studies, with the vast majority demonstrating rigorous design and implementation. Fewer than 20% of studies scored 60 or below, suggesting only a small proportion with notable methodological limitations. Overall, the potential bias was minimal, supporting the reliability and validity of our findings. To enhance transparency, we have incorporated additional sentences into the Result and Discussion sections to strengthen the reporting of these outcomes.

Result section, page 23 line 7-15

“For these qualitative studies, a perfect score (100) indicates that a paper has met all the quality criteria. Such excellence is reflected in a study design that is fully aligned with the research questions and supported by data collection methods that address those questions comprehensively. These papers also demonstrate strong analytical integrity, ensuring that findings arise directly from the evidence and that all interpretations are substantiated by the data. Moreover, they maintain a coherent and logical connection between the data sources, data collection processes, analytical procedures, and final interpretations. Only two studies were rated as having moderate quality (score = 80), as they provided insufficient information to fully support their reported findings.”

Result section, page 23 line 17-23

“Meanwhile, two quantitative non-randomized studies received a score of 60. Scores in this range generally reflect limited clarity and inadequate explanation of core methodological components. In particular, the absence of detailed information on measurement methods, sampling strategies, and population coverage made it impossible to fully assess the rigor of these studies. These omissions may also limit the generalizability of their findings.”

Discussion section, page 39 line 271-277

“Data sources varied substantially across the studies, providing limited information on the extent to which the working population was covered and demonstrating considerable methodological heterogeneity (e.g., differences in reporting systems, types of occupational diseases, and characteristics of the populations studied). This variation restricted our ability to synthesize the data. Nonetheless, given that most of the studies were of high quality (85.9% scored above 80 out of 100), the findings of this review are considered robust and can be interpreted with a high degree of confidence.”

3. Lastly, please change the title of your manuscript, as the word "trends" does not correspond to the content, as you did not assess trends over time. An appropriate title could be: "Global reporting and under-reporting of occupational diseases: a systematic review".

Response:

We have changed the title accordingly.

---

## [Editor Report · Decision Letter 3]

5 Mar 2026

Global reporting and underreporting of occupational diseases: a systematic review

PONE-D-24-50153R3

Dear Dr. Khoe,

We’re pleased to inform you that your manuscript has been judged scientifically suitable for publication and will be formally accepted for publication once it meets all outstanding technical requirements.

Kind regards,

James Mockridge

Staff Editor

PLOS One

---

## [Editor Report · Acceptance letter]

PONE-D-24-50153R3

PLOS One

Dear Dr. Khoe,

I'm pleased to inform you that your manuscript has been deemed suitable for publication in PLOS One. Congratulations! Your manuscript is now being handed over to our production team.

Kind regards,

on behalf of

Dr James Mockridge

Staff Editor

PLOS One